# Plan-Property Dependencies are Useful: A User Study

**Rebecca Eifler,**[1] **Martim Brandao,**[2] **Amanda Coles,**[2] **Jeremy Frank,**[3] **Jörg Hoffmann**[1]

[1]Saarland University, Saarland Informatics Campus, Germany; {eifler, hoffmann}@cs.uni-saarland.de
[2]King's College London, UK; {amanda.coles, martim.brandao}@kcl.ac.uk
[3]NASA Ames Research Center, Mountain View, CA, USA; Jeremy.D.Frank@nasa.gov

## Abstract

The trade-offs between different desirable plan properties –
e. g. PDDL temporal plan preferences – are often difficult to
understand. Recent work proposes to address this by iterative
planning with explanations elucidating the dependencies be-
tween such plan properties. Users can ask questions of the
form "Why does the plan you suggest not satisfy property
$p$?", which are answered by "Because then we would have to
forego $q$" where $\neg q$ is entailed by $p$ in plan space. It has been
shown that such plan-property dependencies can be computed
reasonably efficiently. But is this form of explanation actu-
ally useful for users? We contribute a user study evaluating
that question. We design use cases from three domains and
run a large user study ($N = 40$ for each domain, ca. 40 min-
utes work time per user and domain) on the internet platform
Prolific. Comparing users with vs. without access to the ex-
planations, we find that the explanations tend to enable users
to identify better trade-offs between the plan properties, indi-
cating an improved understanding of the task.

## 1 Introduction

Explainable AI planning (XAIP) is a growing sub-area of
planning, concerned with variety of ways in which aspects of
plans, planning tasks, planning models can be made under-
standable to users (e. g.(Göbelbecker et al. 2010; Seegebarth
et al. 2012; Fox, Long, and Magazzeni 2017; Chakraborti
et al. 2017; Behnke et al. 2019; Sreedharan et al. 2019b,a;
Chakraborti and Kambhampati 2019; Krarup et al. 2019;
Sreedharan, Srivastava, and Kambhampati 2020)). We refer
the reader to (Chakraborti et al. 2019a) for a survey.

In this work, we are concerned with a particular form
of XAIP, proposed recently by Eifler et al. (2020a; 2020b)
(henceforth *Eif20*), that addresses dependencies between de-
sirable *plan properties*. The targeted context are application
scenarios as described by Smith (2012), where these plan
properties are partially conflicting and where one or multiple
users, potentially with conflicting interests, need to make up
their mind on what the best trade-of is. For example, a prop-
erty initially perceived to be important may be re-evaluated
if it turns out to be a bottleneck excluding many other prop-
erties. In such a setting, one-shot optimal planning with fixed

utiliy values does not make sense. Instead, users need to un-
derstand the conflicts to converge to a utility function or oth-
erwise acceptable trade-off. Eif20 introduce an explanation
framework supporting just such understanding, through user
questions about a given plan $\pi$: Why is $\pi$ a good plan? Why
does $\pi$ not satisfy a property I care about? What alternatives
are there? Such questions can be used in a variety of set-
tings, e. g. to build trust in a plan $\pi$ suggested by any plan-
generation method. Like Eif20, here we assume the setting
proposed by Smith (2012), namely an *iterative planning* pro-
cess where users iteratively refine example plans $\pi$.

Plan properties in Eif20's framework are LTLf (De Gi-
acomo, De Masellis, and Montali 2014) formulas over ac-
tions and facts. The approach assumes a fixed set $P$ of such
properties, and computes *dependencies* between those, in the
form of entailments in plan space. Namely, given $p, q \in P$,
$p$ entails $\neg q$ in plan space if all plans that satisfy $p$ do not
satisfy $q$. Given this information, if the user asks "Why does
$\pi$ not satisfy $p$?", the method answers "Because if it did we
would have to forego $q$".[1]

Eif20 show that the set of all plan-property dependencies
can be computed reasonably efficiently. But are the resulting
explanations actually useful to users? Do they help users in
understanding the trade-offs and incompatibilities between
plan properties? We contribute a large user study evaluating
that question, in terms of test-person performance in several
case studies on iterative planning.

The ideal user study would be run with real-life domain
experts tackling difficult plan-preference trade-offs as part
of their profession. However, experts are scarce and difficult
to get access to compared to university students and crowd
workers. In this paper we use crowd users since this has the
advantage of reaching large user numbers $N$. We specifi-
cally use Prolific[2] (Palan and Schitter 2018), which is more
suited for longer user studies, with complex tasks, than for
example Amazon Mechanical Turk. As we describe later, we
use this functionality to re-invite users for follow-up studies
and build up expertise in the use of the explanation tool. We

---

[1]This is a form of contrastive explanation (Miller 2019). Re-
lated approaches (Smith 2012; Fox, Long, and Magazzeni 2017;
Cashmore et al. 2019; Krarup et al. 2019) compare $\pi$ to an al-
ternative plan $\pi'$ that satisfies $p$; here, the answer consists of the
properties $q$ shared by all possible $\pi'$.

[2]https://www.prolific.co/

run the user study using the web-based platform for iterative planning by Eifler and Hoffmann (2020).

We design use cases suited to this setting, encoding preference trade-offs that are complex enough to render plan-property dependencies non-trivial to see, while being easy enough to be solved within the limited time span crowd workers are willing to invest (less than an hour). We run our study across three different planning domains, carefully chosen and adapted for our evaluation purposes. In particular, we introduce a new domain "Parent's Afternoon" encoding the need to drive children to sports events etc. while also taking care of things like shopping, all under timing constraints stemming from pick-up/drop-off/opening times. Our rationale is that this kind of problem is familiar to many people and thus somewhat alleviates the lack of real expert users. Our two other domains are variants of transportation with fuel consumption (Transport, adapted from IPC NoMystery) and Mars-rover data collection and communication (Rovers, adapted from IPC Rovers). The three domains explore different forms of relevant problem structure, in terms of different sources of conflicts between plan properties, namely competition for resources (Transport), time windows/deadlines (Parents Afternoon), and both (Rovers). While Transport and Rovers are not familiar to everyday users, the underlying reasons for conflicts arguably are natural and easy to understand.

An important design decision in our user study pertains to user motivation. In our targeted application scenarios, this motivation is intrinsic – experts will work towards understanding the trade-offs as best possible. In a user study however, test persons have no intrinsic motivation. We therefore opted to task them with additive-reward maximization, assigning a fixed utility to each plan property. We link that objective to payment via a bonus growing with the objective value achieved, thus providing a strong incentive to find good plans. We run our user study with $N = 40$ test persons on each domain, split into two equal-size groups having vs. not having access to Eif20's explanation facilities. Our results show that users with access to explanations tend to identify better trade-offs between the plan properties, in particular re-invited users who have already built up expertise with the tool. This provides evidence that the explanations improve users' understanding of conflicts as intended.

## 2 Background

In what follows we briefly provide the necessary background regarding our planning framework and Eif20's approach, informally only as the technical details are not necessary to understand our user study and contribution. We then describe the web-based platform for iterative planning which we use for our study (Eifler and Hoffmann 2020).

### 2.1 Planning Framework and Eif20

Our investigation is placed in the context of oversubscription planning (OSP) (Smith 2004; Domshlak and Mirkis 2015). An OSP planning task (short: OSP task) $\tau$ defines an initial state $I$ and actions $A$ over a set of state variables (or Boolean propositions/facts) as usual in AI planning. We as-

sume non-negative action costs given by an action-cost function $c : A \to \mathbb{R}_0^+$. There is a bound $b$ on the action cost we are allowed to incur, i.e., action sequences whose summed up cost exceeds $b$ are not allowed. There is a set $G^{\text{hard}}$ of hard goals (state-variable values/Boolean facts) that must be achieved; and a set $G^{\text{soft}}$ of soft goals that are of interest but are not mandatory.

In contrast to standard OSP frameworks, we do not define a utility over $G^{\text{soft}}$. Instead, $G^{\text{soft}}$ represents a set of plan properties, specifically LTL plan-preference formulas compiled into (soft-)goal facts (Baier and McIlraith 2006; Edelkamp 2006; Eifler et al. 2020b). The explanation facility by Eif20 that we evaluate in our user study identifies dependencies between these plan properties. The targeted applications are ones where one-shot optimization over pre-fixed utilities is not desirable, and users instead want to understand how their preferences interact with each other.

The "dependencies" between plan properties here are defined in terms of plan-space entailment. Denote by $\Pi$ the set of plans for the OSP task $\tau$. Say that $\pi \in \Pi$ **satisfies** a formula $\phi$ over $G^{\text{soft}}$, written $\pi \models \phi$, if $\phi$ evaluates to true in the end state of $\pi$. Then $\phi$ $\Pi$**-entails** $\psi$, written $\Pi \models \phi \Rightarrow \psi$ if every $\pi$ with $\pi \models \phi$ also satisfies $\pi \models \psi$. In other words, the plan space $\Pi$ takes the role of a knowledge base.

Eif20 introduce algorithms that effectively identify all **exclusion dependencies** of the form $\Pi \models \bigwedge_{g \in X} g \Rightarrow \neg \bigwedge_{g \in Y} g$ where $X, Y \subseteq G^{\text{soft}}$. Such a dependency holds if all action sequences in $\tau$ whose cost is $\leq b$, that achieve $G^{\text{hard}}$, and that achieve all $g \in X$, do not achieve at least one $g \in Y$. Eif20 observe that the strongest dependencies of this kind correspond exactly to **minimal unsolvable goal subsets (MUGS)** $X \cup Y = G \subseteq G^{\text{soft}}$ where $G$ cannot be achieved but every $G' \subsetneq G$ can. Their algorithms thus compute all MUGS, as an offline process which prepares the answers to all possible user questions.

### 2.2 Web-Based Iterative Planning Tool

We use Eifler et al. (2020) web-based tool for iterative planning with eXplanation through Plan Properties, called **XPP**. XPP runs in standard browsers and is thus ideally suited for web-based user studies, not requiring any installation effort on the part of the test persons.

Study designers in XPP set up the domain and OSP task, including in particular the set of plan properties. In the interface to layperson users, XPP supports natural language descriptions of the plan properties, and visual depiction of the OSP task as an image.

The iterative planning process in XPP is driven by the user's selection of goals to enforce. Each iteration proceeds as follows: 1. the user ticks a subset $G_{\text{enf}} \subseteq G^{\text{soft}}$ to be enforced in the plan; 2. XPP calls a planning server to compute a new plan that satisfies $G^{\text{hard}} \cup G_{\text{enf}}$; 3. the user can optionally ask questions about that plan; 4. the user selects a new $G_{\text{enf}}$, and the process iterates.

Importantly, the planner in step 2. is *not* an OSP planner, but simply a satisficing planner whose only objective is to achieve $G^{\text{hard}} \cup G_{\text{enf}}$. This is the canonical choice in target

scenarios where no pre-fixed utilities over $G^{\text{soft}}$ are given.[3]

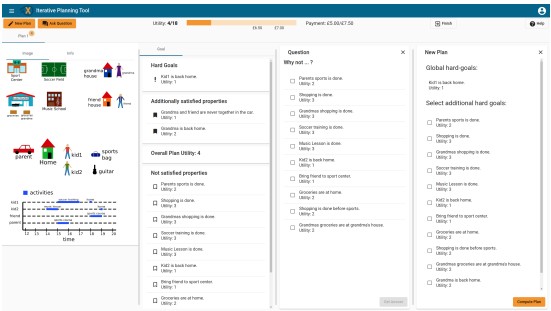

Figure 1: Screenshot of the XPP tool from the user's (test persons's) perspective.

Figure 1 shows a screenshot from the user's (test persons's) perspective. The interface is divided into 4 columns. In the first column, the user has access to the image and textual description of the OSP task. $G_{\text{enf}}$ of the currently selected plan is listed in the second column. The interface for asking questions and selecting $G_{\text{enf}}$ for the next plan is contained in the third and fourth columns, respectively.

## 2.3 Explanations in the Tool

The basic idea in Eif20's method, as already indicated, is to facilitate question – answer pairs of the form "Why does $\pi$ not satisfy $p$?" – "Because if it did we would have to forego $q$". But it's actually a bit more complicated than that. First, $p$ may have many entailments so the answer is not necessarily a single $q$. Second, there are special cases where $p$ can be made true without invalidating any $q$ currently true; and cases where $\pi$ does not exist (the selected $G_{\text{enf}} \subseteq G^{\text{soft}}$ is unsolvable). Altogether, this leads to the following three cases and forms of explanation, in XPP as well as our user studies:

(A) Question: "Why does $\pi$ not satisfy $p$?"

Situation: $G_{\text{enf}}$ solvable, current plan $\pi$ satisfies $G_{\text{sat}} \subseteq G^{\text{soft}}$ with $G_{\text{enf}} \subseteq G_{\text{sat}}, p \notin G_{\text{sat}}$; $G_{\text{enf}} \cup \{p\}$ unsolvable.

Answer: List of sets $G_i$ where there exists a MUGS $G$ with $p \in G$ and $G_i = (G \setminus \{p\}) \cap G_{\text{sat}}$.

Example: "Why can I not go shopping before sports?" – "Because then you can not bring your kid to the music lesson."

(B) Question: "Why does $\pi$ not satisfy $p$?"

Situation: $G_{\text{enf}}$ solvable, current plan $\pi$ satisfies $G_{\text{sat}} \subseteq G^{\text{soft}}$ with $G_{\text{enf}} \subseteq G_{\text{sat}}, p \notin G_{\text{sat}}$; but $G_{\text{enf}} \cup \{p\}$ is solvable (i. e., $p$ was not enforced but could be).

Answer: "Actually yes, we can satisfy $p$ in addition to the currently enforced properties."

Example: "Why can I not bring my friend to the sport center?" – "Actually yes, you can do that in addition to bringing your kid to the music school."

(C) Question: "Why is there no plan?"

Situation: $G_{\text{enf}}$ unsolvable.

Answer: List of MUGS $G$ where $G \subseteq G_{\text{enf}}$.

Example: "Because you can not bring your friend to the sport center, your kid to the music school, and grandma to the supermarket."

# 3 Case Study Design

The user study design space begins with the choice of planning domains and instances. We next outline our design rationales as well as the concrete OSP tasks we use.

## 3.1 Domain Choice and Design Rationales

Folklore wisdom about evaluation in general AI planning is that there should be many benchmark domains covering different forms of problem structure as relevant to the approach evaluated. For user studies though, "many" is constrained by the effort involved in setting up and running such a study. We implemented user studies in three different domains, which is substantial by comparison to other user studies, of which many use 1 domain (Chakraborti et al. 2019b; Chakraborti and Kambhampati 2019; Sreedharan et al. 2019a, 2020; Lindsay et al. 2020; Das, Banerjee, and Chernova 2021) or 2 domains (Sreedharan et al. 2019b; Sreedharan, Srivastava, and Kambhampati 2020).

As already discussed, the most relevant "problem structure" in our context arguably is the source of conflicts between different soft-goal preferences. Our new domain Parents Afternoon encodes overconstrained afternoon tasks of a busy parent, which involves driving children to sports events, shopping, etc. The source of conflicts are pick-up/drop-off/opening times (modeled in classical planning through sequential time stamps encoded as part of the state).

Our two other domains, Transport and Rovers, are variations of the IPC domains NoMystery and Rovers. Transport encodes transportation of packages on a roadmap with fuel consumption, so that soft goals compete for the same consumed resource. Competition for resources is a ubiquitous source of conflict – in everyday life, consider money for instance – so this structure also is natural, and should be familiar to lay users to a certain degree.

Rovers encodes data collection and transmission on Mars, constrained by both resource consumption and timing constraints and thus combining the two conflict sources above. This problem is of course not familiar to layperson users at all[4], but again the conflict-inducing structure is natural and easy to appreciate. NASA employees would be the perfect test-person group here, but for administrative reasons this is not possible. We have limited the complexity to a level that can be handled by a layperson.

Another issue that requires careful attention is the complexity of the domain instance, i. e., the OSP task. As we found in some preliminary tests, an overly complex instance discourages test persons so that they give up without trying, resulting in unusable data. Also, the responsiveness of

---

[3]That said, iterative planning with preliminary/experimental utilities can make sense, and in this case OSP planning over those makes sense in each iteration. Exploring this is a possibility for future work on iterative planning, which however is not our core focus here.

[4]Mars-men aside, who at present cannot register as Prolific users (to our awareness).

the tool has a high impact on the frustration level of the test persons; new plans are computed online during iterative planning (it would be infeasible to pre-compute plans for all solvable subsets of $G^{\text{soft}}$), and this process should be fast; we used 30 seconds as our threshold. Furthermore, the plan properties themselves – LTL formulas, in general – must be easy to understand for laypersons. In particular, it must be easy to see whether or not a given plan property is satisfied by the current plan. For this reason we did not make use of complex temporal dependencies. Finally, the OSP task must be simple enough to be addressed in the limited time span Prolific users will spend on a study.

On the other hand, the instances must be complex enough to contain non-trivial exclusion dependencies. Specifically, MUGS of size 2 tend to be easy to identify and remember, so we designed our tasks to mostly feature larger MUGS, incorporating complex dependencies and thus a challenging plan space. Along similar lines, we tried to avoid "bottleneck" plan properties appearing in a large fraction of MUGS.

## 3.2 OSP Task Design for our User Studies

**Parent's Afternoon**  Our new domain models a parent driving her family members to their activities, like soccer training, while taking care of tasks like grocery shopping. In doing so, the time constraints imposed by the opening times of the activities must be respected. The opening time is the latest time point where the person performing the activity and her equipment must be at the matching location. Further constraints are the overall available time and the capacity of the car. The instance we designed is depicted in Figure 2. We defined 13 plan properties, also given in Figure 2, reflecting doing activities and bringing people and items back home. The instance has 25 MUGS (size 2: 3, size 3: 13, size 4: 7, size 5: 11, size 6: 1).

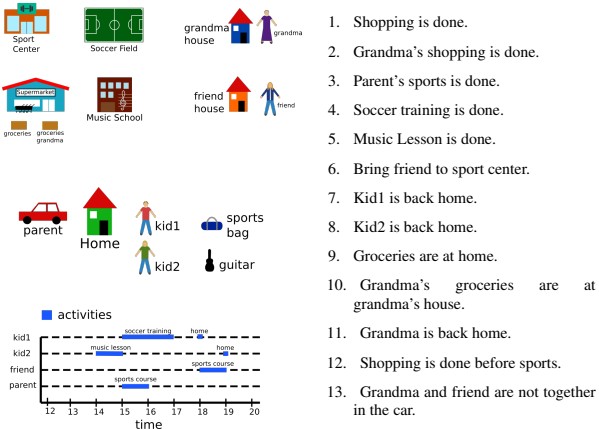

1. Shopping is done.
2. Grandma's shopping is done.
3. Parent's sports is done.
4. Soccer training is done.
5. Music Lesson is done.
6. Bring friend to sport center.
7. Kid1 is back home.
8. Kid2 is back home.
9. Groceries are at home.
10. Grandma's groceries are at grandma's house.
11. Grandma is back home.
12. Shopping is done before sports.
13. Grandma and friend are not together in the car.

Figure 2: Image and plan properties of the parent's afternoon instance used on the user study.

**Transport**  Transport encodes transportation of packages on a roadmap with fuel consumption, similar to IPC No-Mystery but extended with limited truck capacity. The instance we designed has 9 locations arranged in a grid with fuel costs between 1 and 3. There are 2 trucks of capacity 1, which need to deliver 5 packages. The 15 plan properties reflect the delivery of packages, the use or non-use of road connections, locations visits, and and ordering relations between pairs of packages. There are 37 MUGS (size 3: 14, size 4: 7, size 5: 2, size 6: 2).

**Mars Rover**  This domain models a rover performing tasks (images, x-ray images, soil samples) at different target locations and uploading the collected data to a relay satellite. The rover has limited power and storage capacity and there is limited time available for the mission. Each task consumes a type-dependent amount of energy and time. Driving from one location to an other also consumes time and energy. Uploading data to the relay and taking a normal picture is only possible in certain time windows, where the relay in is transmission range and the target is illuminated respectively. We designed an instance with one rover, 4 locations, and 10 tasks (4 images, 2 x-ray images, 4 soil samples). Among the 14 plan properties, in addition to uploading data to the relay, is the order of data uploads. There are 102 MUGS (size 2: 22, size 3: 56, size 4: 4). This instance has about 3 to 4 times more MUGS than the other two instances, but has more MUGS of size 2.

# 4  User Study Design

With the underlying planning benchmarks clarified, we now turn to the user study itself, which also involves many design decisions. We next discuss user motivation, then cover the generaal setup (test-person recruitment, payment, etc.) and experiment workflow.

## 4.1  User Objective

In the application scenarios targeted by Eif20, expert users are working to understand conflicts between preferences and thus converge to a utility function or otherwise acceptable trade-off. But in a non-expert user study, test persons will not have an intrinsic motivation to do so. Hence we give them an objective to pursue, namely additive-reward maximization which is canonical as it is easy to understand for layperson users. We assign a fixed utility to each plan property.[5]

In what follows, keep in mind that this objective is *only in the heads of the test persons*. Neither the satisficing planner called by XPP nor the explanation method take it into account. This is because, in the targeted application scenarios, no such fixed (and simple) objective exists.

We link the user objective to payment via a bonus growing with the objective value (summed-up reward) achieved, thus providing a strong incentive to find good plans (some prior work, e. g. (Chakraborti et al. 2019b), has followed similar schemes). Specifically, the base payment for an overall estimated processing time of 40 minutes is $5\pounds$ (corresponds to recommended hourly wage of $7.50\pounds$ by Prolific). Each test person can receive a bonus payment of up to $2.50\pounds$

---

[5]Fixed utility is a standard form of oversubscription planning, which could be solved optimally using known algorithms (e. g. (Smith 2004; Domshlak and Mirkis 2015; Katz et al. 2019)). Nevertheless, this setup is meaningful for evaluating Eif20's explanation approach, as test persons in our study will need to understand the dependencies between plan properties to perform well.

depending on the achieved utility. The bonus payment is divided into 3 levels, of $1.50£/2.00£/2.50£$ for reaching $60\%/80\%/100\%$ of the maximal utility.

We include a progress bar in XPP, conveying to users what the maximal possible utility is, the fraction of that utility reached so far, and the according bonus payment level reached. We found in preliminary test runs that such explicit information helped to motivate test persons.

## 4.2 User Study Setup

We next give the specifics regarding various parameters of our setup, namely recruitement, performance measurement, and the concluding questionnaire to be filled in by each test person after completing the task.

**Test Person Assignment, Recruitment, & Filtering** To evaluate the effect of Eif20's explanation facility, we divided test persons randomly into groups with vs. without that facility, i.e., with vs. without the option to ask questions. We refer to these two groups as Q+ and Q− respectively.

Familiarity with XPP, the iterative planning process, and the explanation facility is a form of expertise. To leverage this expertise as much as possible, we re-invited test persons to address the remaining planning domains as well.

We fixed the ordering of domains to Transport, Parents Afternoon, Rovers, where Rovers is last as it is hardest (combining both sources of conflicts) while the ordering of Transport and Parent's Afternoon is arbitrary. We waited with domain $i$ until the user study on domain $i − 1$ was completed, so as to maximize the number of re-invited test persons (data on that number will be given in Section 5). We fixed each test person's assignment to the Q−/Q+ group across all domains, to obtain consistent streams of test persons becoming increasingly familiar with either of the two tool variants, and as distributing re-invited users across both variants would have resulted in too many different subgroups for a meaningful analysis.

We used the test person recruitment facilities of Prolific (Palan and Schitter 2018). We applied several filters on test persons to obtain meaningful results. First, we required fluency in English, and an "accepting score" of $> 50\%$, i.e., $50\%$ of the each test person's previous submissions in Prolific must have been accepted by the respective study organizers. Second, to filter out test persons who did not meaningfully process the user study, they had to pass a few simple sanity-test questions about the processed task. Finally, in the Q+ group, we filtered out those test persons who did not actually use the explanation facility, i.e., who did not ask any questions.[6] For each domain, we kept running the study until we had 20 test persons for each of Q+ and Q−, to a total of $N = 40$ (once 20 was reached for either of Q+ or Q−, we assigned every user to the respective other group).

**Performance Recording** We stored data allowing reconstruction of the entire iterative planning process. For every

---

[6]This is a possible effect of the tool environment complexity in conjunction with the limited time test persons spent. Among test persons using the tool for the first time, the ratio of such drop-outs was 25%. Among re-invited users, there were no drop-outs.

plan produced, this data includes the enforced goals $G_{enf}$ and the plan utility. For every question asked, we store the relevant parameters, i.e., the referenced plan $\pi$ and plan-property $p$ for question types (A) and (B). All these records are associated with timestamps. For a fine-grained analysis of test person strategies, we furthermore record the time spent in each part of the tool interface.

**Questionnaire** In addition to these performance measurements, we included a questionnaire for subjective measures. We use a Likert scale from 1 to 7 to measure the test person's opinions. The questions are listed in Table 2, which is included in our results analysis (Section 5.4) for ease of reading.

Beyond these fixed-answer questions, we furthermore included free-text questions, targeted at qualitatively assessing the presentation and usefulness of explanations in the proposed setting. The Q+ test persons were asked for comments about the structure and presentation of the questions and explanations. The Q− test persons were asked to list questions that would have helped them to solve the task.

## 4.3 Experiment Workflow

Each experiment, i.e. each test-person run addressing the OSP task from one of our domains, proceeded according to the following workflow:

1. **Textual domain description**: A general description of the domain is given. This includes an explanation of the possible goals and more complex plan properties and how they vary in utility. The constraints present in the domain are highlighted and their impact on the satisfiability of the plan properties is addressed.

2. **Textual tool description**: The test persons are introduced to the iterative approach of the tool. Their goal of finding a plan with the maximum utility is emphasized. An instruction manual for the tool, accessible at all times, is provided. For Q+, the questions facility is explained.

3. **Familiarization with tool through introductory instance**: Given the complexity of the task and tool, the test person is familiarized with the domain and tool through a small introductory instance. The instance is described by an image with accompanying explanation text. The image and explanation text can be accessed by the user throughout the study. Test persons must compute at least one plan, and in the Q+ group must ask at least one question before proceeding to the next step.

4. **Planning for evaluation instance**: The test person processes the evaluation instance (as described in Section 3). Again the instance is described by an image with accompanying explanation text, accessible throughout the study. The test person can exit the task at any time, in particular without reaching maximal utility.

5. **Questionnaire**: Finally the test person has to answer the questionnaire.

# 5 User Study Results

We now evaluate the results of our user study. First we briefly give the numbers of test persons in each category.

Then we present our main results, regarding the impact of the explanation facility on performance, in terms of utility achieved over time. We follow this up with an analysis of tool experience, comparing new vs. re-invited test persons, in the Rovers task which is particularly challenging due to its aforementioned complexity. We finally evaluate the questionnaire results, in terms of a statistical analysis of the Likert scale answers, and in terms of a thematic analysis of free-text answers regarding user strategies, criticisms, and desired explanation facilities.

## 5.1 Test Person Statistics

Table 1 shows the data regarding new and re-invited test persons per domain.

| | Transport | | Parent's A. | | Rovers | |
|---|---|---|---|---|---|---|
| | Q− | Q+ | Q− | Q+ | Q− | Q+ |
| new | 22 | 29 | 4 | 6 | 10 | 15 |
| filtered out | -2 | -9 | 0 | 0 | 0 | -2 |
| re-invited | – | – | 16 | 14 | 10 | 7 |
| filtered out | – | – | 0 | 0 | 0 | 0 |
| $\sum$ | 20 | 20 | 20 | 20 | 20 | 20 |

Table 1: Distribution of new, re-invited, and filtered-out test persons per domain and group.

In total, 36 (50) individuals participated as Q− (Q+) test persons in our study, of which 2 (11) were filtered out. The filtered-out Q− users did not meaningfully complete the study, while the 11 filtered-out Q+ users did not ask questions. Note that the latter happend primarily in Transport, and did not happen at all among re-invited users. About 75% of the test persons participated in two of or domains, and 50% of the test persons participated in all three domains.

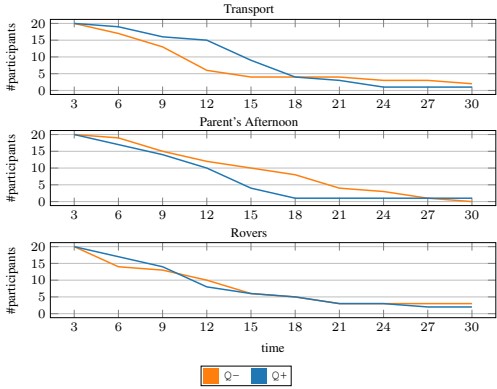

Figure 3: Test person retainance over processing time.

Figure 3 shows the number of test persons as a function of processing time, i.e., the time spent addressing the evaluation task. Despite the performance bonus incentive, that number decreases rapidly over time. This is caused by task complexity. A few persons give up quickly, not trying to get a bonus. Many others leave after reaching 60% utility and thus the base bonus; reaching higher boni requires substan-

tial work and a much larger time investment, as we shall see in the next sub-section.

## 5.2 Performance

Figure 4 shows our primary evaluation of explanation usefulness: user performance for Q− vs, Q+, as a function of processing time.

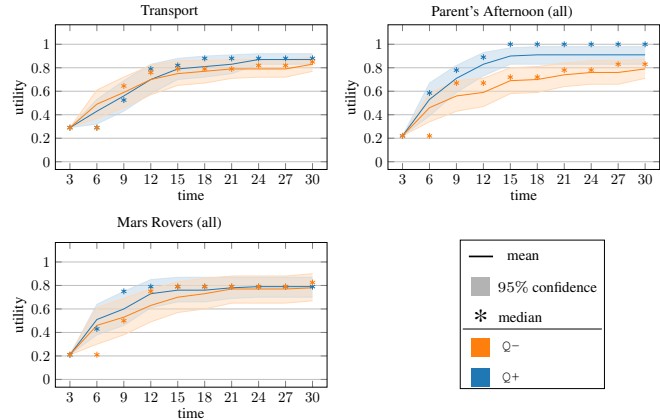

Figure 4: Performance over processing time: x-axis time in min; y-axis maximal achieved plan utility until that time.

The performance advantage of Q+ over Q− is most striking in Parent's Afternoon. In Transport and Rovers, the advantage is also visible, but is not as pronounced. Q+ has higher means and medians in Transport for $t > 12$min, and in Rovers for $t < 21$min.[7] For Rovers, we will see in the next part of our evaluation (Section 5.3) that domain complexity is a major factor: re-invited users perform significantly better than new ones in Rovers, and the performance advantage of Q+ over Q− is more pronounced.

One reason for the differences across domains may be that Parent's Afternoon is most intuitive, and most familiar to laypersons. We believe that the primary reason is task complexity though. Given the exponential nature of the underlying structures (state space size, number of plan-property subsets), the transition from easy tasks to extremely hard tasks is rapid. At the same time, crowd workers have little expertise and are making small time investments only. So our OSP task design (cf. Section 3.2) was walking a fine line between tasks where high utility can be achieved easily even without explanations, vs. ones where that is very hard for our test persons even with explanations. Transport somewhat tends to the former category (60% utility for Q− after 6 minutes), Rovers somewhat tends to the latter (only 80% utility for both groups after 30 minutes). Parent's Afternoon seems to balance that knife edge best, of our three use cases.

Note that these observations pertain to artefacts of the user study setting. In practice, expert users will invest substantial effort to understand preference conflicts in hard tasks. Our

---

[7]In all domains, at $t = 6$ the median for Q− is much lower than the mean, indicating that the high Q− mean utilities at the beginning are due to a few exceptionally well-performing test persons.

results in all three domains indicate that Eif20's explanation facility can help with that.

Let us briefly shed light on user behavior and performance within Q+. Consider Figure 5.

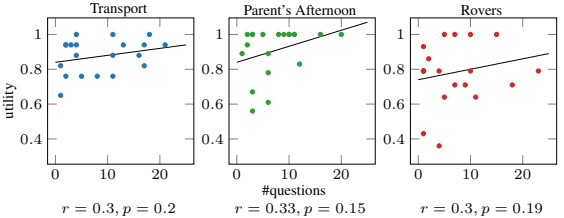

Figure 5: Number of questions asked vs. performance. Linear approximation minimizes squared error; $r$ and $p$ value of Pearson correlation coefficient test.

As the linear approximations indicate, the trend in performance is somewhat upwards for users that ask more questions. Observe also the "lower rim" of each scatter plot: the worst utility achieved by users who asked $x$ questions grows consistently in each domain. This indicates that intensive interaction with the explanation facility leads to a reduced risk of coming up with a bad-quality trade-off.

## 5.3 New vs. Re-invited Users in Rovers

Figure 6 compares the results of new and re-invited test persons in Rovers, thus evaluating the impact of tool experience in this challenging task.

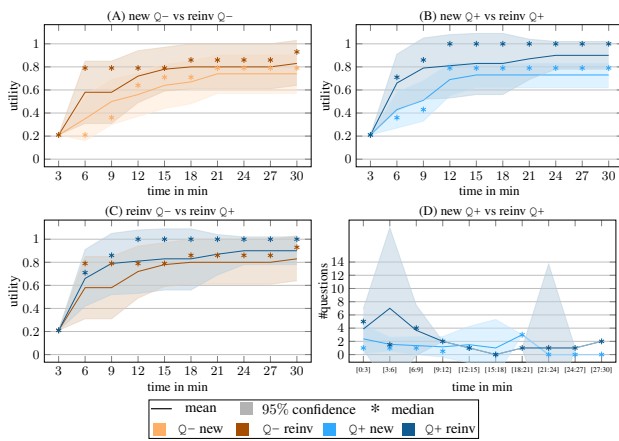

Figure 6: Performance (a)–(c) and #questions (d) over processing time in Rovers domain. (a) Q− new vs. re-invited; (b) and (d) Q+ new vs. re-invited; (c) re-invited Q+ vs. Q−.

As Figure 6 (a) and (b) show, re-invited test persons perform consistently better than new users, for both Q− (a) and Q+ (b), strongly indicating the need for tool expertise in this domain. Such expertise also results in a larger advantage of Q+ over Q− users, as seen in Figure 6 (c) compared to the data for Rovers in Figure 4. Figure 6 (d) shows that the increased tool experience is also reflected in a tendency to ask more questions at specific points in the process.

| | | Question | Likert scale labels (1) | (7) |
|---|---|---|---|---|
| Q− and Q+ | Q1 | How difficult was the task for you? | very easy | very difficult |
| | Q2 | How satisfied are you with your achieved result. | not satisfied | very satisfied |
| | Q3 | How confident are you, that you know which plans are possible/which properties can be achieved together? | not confident | very confident |
| only Q+ | Q4 | The possibility to ask questions helped me. | don't help at all | very helpful |
| | Q5 | The possibility to ask questions reduced the level of difficulty. | not at all | much easier |
| | Q6 | The questions helped me to find better plans. | don't help at all | very helpful |
| | Q7 | The questions helped me, when I wanted to improve a plan. | don't help at all | very helpful |
| | Q8 | The questions helped me, when the selections of properties was unsolvable. | don't help at all | very helpful |
| | Q9 | The questions helped me to understand which plans are possible/which properties can be achieved together. | don't help at all | very helpful |

Table 2: Questionnaire: Likert scale questions

## 5.4 Questionnaire Evaluation

We now analyze the answers to the user questionnaire for subjective measures. Table 2 shows the questions, which consist of two groups, each posing several questions targeting the same measurement in different formulations: (top) comparison of task satisfaction between Q+ vs. Q−users; (bottom) helpfulness of explanation facility for Q+ users. Figure 7 gives data for one question from each group, namely Q2 and Q4. We used the *Student's t-test* to evaluate statistical significance of the difference between means.

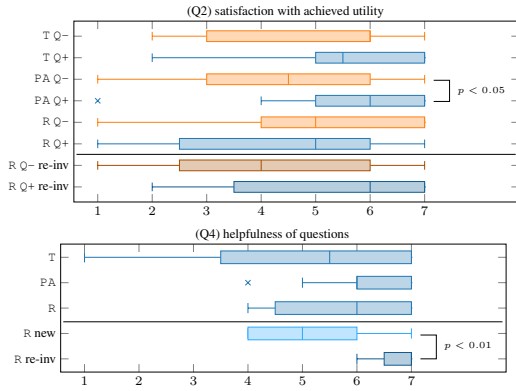

Figure 7: Analysis of questions Q2 and Q4 of Table 2. Abbreviations: Transport T, Parent's Afternoon PA, Rovers R.

Q2 turned out to receive the most interesting responses in the first group of questions (Q1 answers exhibited little variance; Q3 only slightly more, with small advantages for Q+). As Figure 7 (top) shows, subjective user satisfaction tends to be higher for Q+ users. The single exception is Rovers measured over all users, which is due to the aforementioned complexity of this domain; for experienced re-invited users only, the picture is similar to the other two domains.

In the second group of questions, the answers to all questions were quite similar, and Figure 7 (bottom) shows data for Q4 as a representative (and most generically formulated) example. On average, the subjective helpfulness of explanations is rated $\geq 5.5$ for all domains. While for Transport the variance is large, for Parent's Afternoon the explanations are ranked to be (very) helpful by almost all test persons. In

Rovers, there is yet again a significant difference between new and re-invited users, indicating that explanation help-fulness increases with experience.

## 5.5 Qualitative Analysis

Study participants also provided free-text answers where they (a) described their problem solving strategy (Q+ and Q−); (b) provided criticisms of the explanation facility (Q+); (c) elaborated on the kind of explanations they would like to have (Q−). We conducted a thematic analysis of the replies to identify common themes in these answers. We summarize our findings in the order (a) – (c). The answers to (c) also resulted in some criticisms of the tool and planning process, which we describe in what follows along with (b).

**(a) Problem solving strategies.** A common problem solving strategy in both groups was to start by attempting to satisfy highest-utility properties first, before adding lower-utility properties as goals. In addition to this, many (7) users in the Q+group explicitly said they used the question-asking interface to better understand property conflicts: users said they used the explanations to "help guide in what to change", "to see which goals did not work together", and to help deduce "the most frequently conflicting goals". One user said they only used the question-asking functionality as a last resort when they were "lost", and another user purposefully avoided asking any questions because they are "not a person who would like to ask questions".

**(b) Criticisms by Q+ and Q− users.** A common theme here was visualization. Multiple users said they would like to see color-highlights overlaid on the task image, in order to "get a visual representation of what is conflicting", and make explanations more "readable". Visualization was also a popular request in the Q−group: the ability to "draw" plans (3 users); plan animation (4 users); visualization of plans and plan-utility (3 users), for example to see how the "timeline is occupied as I go and select tasks, as well as to see the most time-efficient tasks". It can certainly be concluded that visualization may be a useful topic for future research in the plan-property dependency explanation context.

Some users (3) did not fully understand the meaning of MUGS, asking whether satisfying one property excludes just one, or all, of the other properties (it's the latter as MUGS are minimal). This could probably be improved by a different wording/more explanation in the XPP tool.

A final common theme (4 users) was that some plans took "a long time to compute". In rare cases, this led users to believe the planner got "stuck". Real-time planning methods might thus be of some benefit in iterative planning, though experts in actual applications will presumably be both, better informed about planner behavior and more willing to wait.

**(c) Explanations Q− users would like to have.** The major kinds of explanations Q− users suggested (in addition to visualization, cf. above) can be roughly categorized into four groups: plan-property dependencies; reasons for such dependencies; questions about specific actions in the plan; and planning model clarifications.

The first of these was common (7), mentioning a desire to know which "properties" and property "combinations" are in conflict. This underscores Eif20's approach. There was a tendency to desire seeing all such dependencies (all MUGS), which is unwieldy even in our small study examples. Sophisticated filtering and/or visualization would be required.

One user expressed interest in understanding why a set of properties cannot "work together", while another user requested explanations for why certain properties have many incompatibilities (e. g. "why are so many goals incompatible with taking grandma shopping?"). These questions point to an interesting issue for future work, into *deeper "why" questions* analyzing the reasons for conflicts.

Three users mentioned the desire to ask questions regarding specific actions in a plan, using counterfactuals such as "why truck 2 must go to the Bank?" (as opposed to not going to the Bank), or "why is this the first move made?" (instead of any other move). MUGS could be one potential answer to such questions, but other XAIP approaches (e. g. plan-step explanation (Seegebarth et al. 2012) or model reconciliation (Chakraborti et al. 2017)) might also be relevant.

The last kind of question pertained to clarifications of the underlying planning model. This was actually the most frequent kind of question (13 users). For example, users asked whether a package can "be left somewhere that another truck could pick it up", whether a truck can "go back the same way it came", or whether a guitar can be left at the "music lesson". This is presumably largely an artifact of the limited time test persons spent trying to understand the domains. Nevertheless it points to a potential new direction for XAIP, answering questions about whether certain actions, states, or sequences of actions are permitted/possible.

## 6 Conclusion

Eif20 introduced a framework for the explanation of conflicts in OSP tasks, and showed how to compute the required information (the MUGS) reasonably effectively. An evaluation whether this form of explanation is useful for users was missing so far. Our work fills this gap. In a sizeable user study across three domains, we found that the explanations tend to enable users to find better trade-offs, especially in the new domain Parent's Afternoon which is intuitive and familiar to layperson users.

As far as evaluating Eif20's explanation approach goes, we believe that this answers the main questions. Future work pertains primarily to further developments of Eif20's approach, such as visualization of conflicts, and supporting deeper why questions as indicated by some of our test persons. Beyond Eif20's approach, our qualitative analysis of free-text replies furthermore points to a direction that may yet be underdeveloped in XAIP, namely answering questions about the planning task semantics.

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
