# OpenReview forum: "Plan-Property Dependencies are Useful: A User Study"
_icaps-conference.org/ICAPS/2021/Workshop/XAIP — XAIP 2021_

### Official Review · AnonReviewer1 · 2021-06-30
**While the authors did a reasonable job to design a user study, some statistical evaluation on objective tests are needed to complete the evaluations and show the results.**

**Rating:** 7
**Confidence:** 5

**Review:**

This paper conducts a web-based user study evaluating the usefulness of the explanation that addresses dependencies between desirable plan properties. They build their user study based on the work Eif20 that proposes iterative planning with explanation to address plan property dependencies in which these plan properties are partially conflicting where users with possible conflicting interests should find the best tradeoff. So, this paper tried to conduct a study to evaluate whether (1) these explanations are useful to users (2) these explanations can help users to understand the tradeoff between plan properties. Three planning domains are used in the study Parent’s afternoon, Transport with fuel consumption, and Rover. The authors added additive reward maximization associated with bonus payment to make the intrinsic motivation for users to act better and act like experts in the study. And for the explanations, they preprocessed minimal unsolvable goal subsets (MUGs).

Strength
- The paper is well written with detailed information about the study.
- For the study, many aspects have been considered and the authors did a reasonable job for designing the study some of these considerations are:
    - Considering simple OSP tasks to address users’ limited span and threshold for giving up without a try.
    - Considering nontrivial MUGS.
    - Having both objective and subjective evaluation.
    - Evaluating both new and re-invited users.

Concerns and Questions:

1. In the first case of explanations (A) on page 3, if q is more than one would you list all of them or choose some of them to show? If you choose between them, what are the criteria for choosing the one to show in explanation?
2. In the third explanation and question case (C) on page 3, you will list all MUGS while explaining that these goals are unsolvable may not be very convincing to users as in the qualitative analysis some users pointed out the need for a reason for conflict, do you have any alternative for this issue?
3. While you checked if the explanations would be useful to understand the tradeoff, it would be also good to evaluate the effect of different cases of explanation separately, since each case may have different effects on the users, in such cases you would find a better connection between the questions, explanations and the utilities in each case.
4. While considering simple OSP tasks is good for user study, but it might not capture the difficult scenario with real expert users and cannot generalize the usefulness of such explanations for real users.
4. This is my main concern about the evaluations, while you have statistical evaluations for the subjective questions, I believe for the results in the objective evaluation you need to have statistical evaluations to validate the hypotheses of useful explanations and better understanding. For instance, in figures 4 and 5 we can see better utility in Q+ case than Q- case (at least visible in Parent’s afternoon), but even in the parent’s afternoon domain, without having some statistical evaluation such as ANOVA and p-value and having significant results you cannot say the results are better in Q+ case. So, I strongly suggest you to add statistical evaluation such as mixed-design ANOVA to evaluate the utility increase over time in both cases or any statistical evaluation which might be appropriate for the study. Because without them the evaluation is incomplete.

---

### Official Review · AnonReviewer2 · 2021-07-06
**The paper makes a modest contribution to XAIP through analysis of a user study assessing the potential usefulness of a previously proposed algorithm for generating particular types of explanations designed to enable better human decision making.**

**Rating:** 6
**Confidence:** 4

**Review:**

The paper makes a contribution to XAIP with the results and analysis of a user study assessing the potential usefulness of a previously proposed algorithm for generating particular types of explanations designed to enable better human decision making.

The article contains a brief overview of the prior method, describes the three selected test domains, explains the experimental method, then provides summary statistics of the results and some qualitative analysis of free form responses.
The article is overall clearly written and well organized. The topic – user studies involving XAI mechanisms – is important, and understudied. The number of subjects (40 per domain, in a with/without support design) is substantial, and the results are reported in some detail.

Some possible improvements
-	“bottleneck” – not defined
-	In section 3.2, give some concrete examples of the MUGS, not just the counts
-	The three domains involve constraints on resource, constraints on timing, and constraints on both; but from a cognitive complexity/load perspective, it is not obvious (to this reader) that these are importantly distinguished from “generic” constraints in the human reasoning process; this distinction becomes relevant when considering the relative difficulty of the tasks in the three domains – we have the sizes of the MUGS sets (a form of syntactic complexity), but should there be some parallel analysis of the ‘semantic’ complexity?
-	Because the number of re-invited subjects has varied across the domains and conditions, and because it seems there is a demonstrable ‘expertise’ effect, should the cohorts have been deliberately balanced, and the potential training effect have been explicitly considered in sections 5.1 and 5.2?
-	The numbers of subjects form which qualitative assessment was reported in section 5.5 is low, compared to apparent confidence the authors have in the resultant conclusions
-	There are standardized tests of workload, trust, explanation satisfaction (etc) in the literature – some reference to these (or use, where appropriate) would be more appropriate than the questions in Table 2 – where no source of validation was offered.
-	As mode of presentation is known to be a substantive influence on comprehension by human subjects, some of the comments in 5.5(b) apply not only to future work, but perhaps also suggest some interrogation of the generalizability of the reported results; this, in particular, puts some doubt into the claim in the Conclusion that “as far as evaluating Eif20’s explanation approach goes, we believe that this answers the main questions”.

---

### Meta-Review · Area_Chairs · 2021-07-07

**Recommendation:** Accept
**Confidence:** 5

**Metareview:**

Thank you for your submission.

This paper evaluates a previous framework for explainable AI planning through a user study.

The reviewers agree that user studies for evaluating XAI techniques are important and that the study is well designed and of a significant size. They agree that the results are well presented and that overall the paper is well written.

Main points raised by reviewers:
- Are the hypotheses of the user study defined well enough and statistically evaluated?
- Standardised tests for explanation effectiveness and existing work related to metrics for explainable AI.
- Semantic complexity of the domains used.
- Evaluation of different types of question.
- Size of domains, difficulty of LTL formulas.

Finally I would be interested to know if the properties you defined for each domain had an effect on the outcome. Or how easy it would be for users to define properties that would cover a useful coverage of questions one might pose about a domain.

We hope that these reviews will be helpful in developing this work further. Please consider the comments for the camera-ready version. We look forward to your presentation!

---

### Decision · Program_Chairs · 2021-07-08

Accept